# Expression Sampler as a Dynamic Benchmark for Symbolic Regression

**Ioana Marinescu***
Princeton University
`ioanam@princeton.edu`

**Younes Strittmatter***
Brown University
`younes_strittmatter@brown.edu`

**Chad C Williams**
Brown University
`chad_williams@brown.edu`

**Sebastian Musslick**
Osnabrück University, Brown University
`sebastian.musslick@uos.de`

**\*Equal contribution**

## Abstract

Equation discovery, the problem of identifying mathematical expressions from data, has witnessed the emergence of symbolic regression (SR) techniques aided by benchmarking systems like SRbench [1]. However, these systems are limited by their reliance on static expressions and datasets, which, in turn, provides limited insight into the circumstances under which SR algorithms perform well versus fail. To address this issue, we introduce an open-source method[1] for generating comprehensive SR datasets via random sampling of mathematical expressions. This method enables dynamic expression sampling while controlling for various expression characteristics pertaining to expression complexity. The method also allows for using prior information about expression distributions, for example, to simulate expression distributions for a specific scientific domain. Using this dynamic benchmark, we demonstrate that the overall performance of established SR algorithms decreases with expression complexity and provide insight into which equation features are best recovered. Our results suggest that most SR algorithms overestimate the number of expression tree nodes and trigonometric functions and underestimate the number of input variables present in the ground truth.

## 1 Introduction

The automated discovery of mathematical equations plays a major role in accelerating scientific discovery. Symbolic Regression (SR) is a family of data-driven methods designed to discover such mathematical equations [2]–[9]. The backbone of SR research is benchmark datasets that enable comparing different types of methods, exposing their successes and failures in recovering different equations [1], [10]. However, conventional benchmarking practices typically rely on a small static set of equations [1], [2], [11], [12]. These provide no systematic insights into which aspects of equations facilitate or hamper recovery, underscoring the need for a more adaptable benchmarking paradigm. This study presents such a paradigm in the form of an expression sampler capable of controlling for various metrics of expression complexity and prior information about expression distributions.

There are several benchmark datasets for SR methods. Some of them, like the Feynman Symbolic Regression Database (FSReD) [2], are inspired by real-world equations. FSReD consists of 100

---

[1]The method is available as a python package and documented at https://autoresearch.github.io/equation-tree/.

NeurIPS 2023 AI for Science Workshop.

equations based on the Feynman Lectures on Physics [13]–[15] and 20 more challenging (bonus) equations from other physics books [16]–[19]. Udrescu and Tegmark [2] provide data tables for each equation accompanied by tables of the physical units for SR algorithms that can use these units (for example, [20]). Inspired by Hoai, McKay, Essam, *et al.* [21], Keijzer [22], and Johnson [23], Uy, Hoai, O'Neill, *et al.* [11] suggested ten different real-valued SR equations and created the corresponding dataset (Nguyen dataset). They generated each dataset from the same set of equations by randomly sampling 20 - 100 data points. As synthetic benchmarks are not always indicative of real-world scientific discovery, Cranmer [4] introduced EmpiricalBench, consisting of 9 equations discovered from experimental data; Cornelio, Dash, Austel, *et al.* [24] used Kepler's Third Law, Relativistic Time Dilation, and Langmuir's Adsorption Equation. Cava, Orzechowski, Burlacu, *et al.* [1] designed an SR benchmark, named SRBench, and conducted a comprehensive benchmark experiment using existing SR datasets such as FSReD and Ordinary Differential Equation Strogatz repository [12]. In SRBench, SR methods are assessed based on 1) the squared error between estimated and ground truth data and 2) the solution rate corresponding to the percentage of estimated equations that match the ground truth after simplification [25].

Existing approaches to benchmarking SR algorithms suffer from a critical limitation: they rely on a static set of equations. Such static sets provide limited insight into which aspects of equations facilitate or hinder discovery by SR methods. To address these limitations, we introduce a novel approach that offers flexibility in the evaluation process by enabling dynamic sampling of mathematical expressions. Researchers and practitioners can adjust key parameters pertaining to expression complexity, such as expression length, the number of input variables, the number of constants involved, and features specific to the distribution from which expressions are sampled. As such, researchers can tailor the benchmarks to suit the needs of different scientific disciplines or problem domains and investigate the strengths and weaknesses of specific SR algorithms. The ability to control various metrics for expression complexity enables a more systematic diagnosis of which aspects of equations make SR algorithms fail or succeed. Altogether, we make the following contributions:

1. We introduce an expression sampling algorithm to benchmark SR methods systematically. The algorithm allows researchers to control various parameters of expression complexity and tailor generated expressions to a specific scientific domain by incorporating domain-specific priors for mathematical expressions.

2. We compare several SR methods with respect to their ability to recover equations from dynamic datasets and examine the impact of different equation complexity metrics on equation recovery. Our results indicate that various complexity metrics have distinct effects on different SR algorithms.

## 2 Expression Sampler

We introduce an expression sampler designed to streamline the benchmarking of SR algorithms. We begin with outlining the expression format utilized by our sampler—an incomplete binary tree (with unary or binary nodes). Subsequently, we detail the sampling methods that can be used for SR benchmarking.

### 2.1 Expression Tree

Our sampler represents mathematical expressions as incomplete binary trees, with *operators* and *functions* as internal nodes and *features* as leaves (see Figure 1). In this framework, *operators* (e.g., addition, subtraction multiplication) operate on two operants (requiring two child nodes). *Functions* (e.g., cosine, sine, square root) operate on a single operant (requiring a single child node). *Features* devoid of any operant represent either variables or constants (for more details, see Appendix A.1).

An advantage of the aforementioned tree representation is the dissociation between an expression's *structure* and its content. The *structure* is the tree configuration, which can be mapped by a *preorder traversal*, recording the node *depth* (number of edges from the root node) without its value. In *preorder traversal*, the left subtree of each node is evaluated first, beginning with the root and then recursively exhausting each left subtree before moving on to the right subtree. Examples of tree structures are depicted in Figure 1: The expression $x_1 + \frac{3}{x_1}$ depicted in Figure 1A has the structure $(0, 1, 1, 2, 2)$. Similarly, the expression $\sin(x_1 + x_2^{x_3})$ in Figure 1B has a structure of $(0, 1, 2, 2, 3, 3)$,

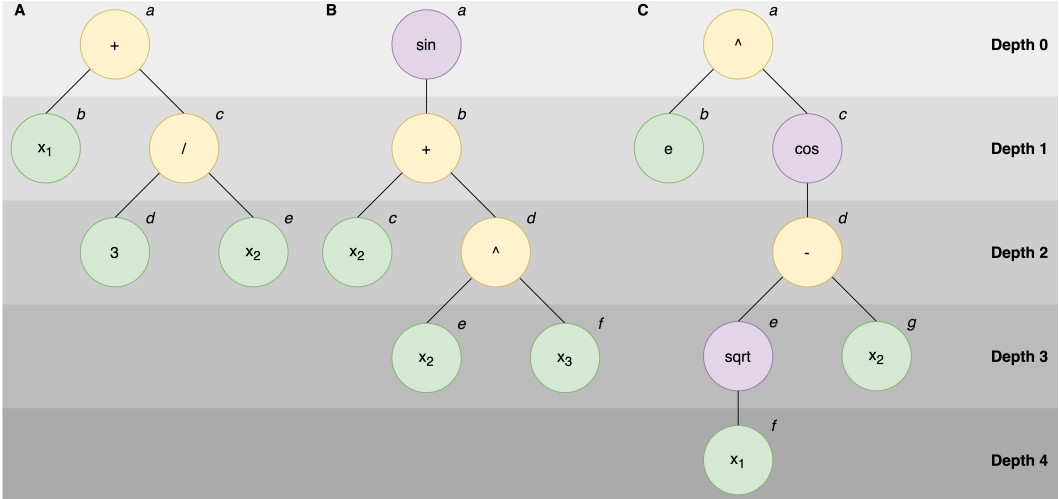

Figure 1: Examples for expressions as incomplete binary trees: Tree (A) represents $x_1 + \frac{3}{x_1}$. Tree (B) represents $sin(x_1 + x_2{}^{x_3})$. Tree (C) represents $e^{cos(\sqrt{x_1} - x_2)}$. Operators (yellow) have two child nodes. Functions (purple) have one child node. Features (green) have no child node. Traversing in preorder (left subtree first; letters next to nodes indicate traversal order) and noting the node depth (number of edges from the root node; indicated on the left side of the figures), we obtain the following structures: tree (A) has the structure $(0, 1, 1, 2, 2)$, tree (B) the structure $(0, 1, 2, 2, 3, 3)$, and tree (C) the structure $(0, 1, 1, 2, 3, 4, 3)$.

while $e^{\cos(\sqrt{x_1} - x_2)}$ in Figure 1C is structured as $(0, 1, 1, 2, 3, 4, 3)$. Different expressions can have the same structure. For example, $x_1 + x_2$ and $x_1 - 1$ have the same structure but different content.

## 2.2 Expression Sampling

We begin by sampling tree structures without any node values. For a given structure, nodes with two children are populated through sampling from operators, while those with one and zero children are sampled from functions and features, respectively. This procedure leaves us with a sampled tree with values in each node, but the tree might not be a valid expression or might be overly complicated. Therefore, after sampling the expression, we employ 'Sympy' [25] to simplify the expression and then convert it to standard conventions (see Appendix A.1). We subsequently perform validity checks, discarding any invalid expressions, such as $\sqrt{-(x^2)}$ or $\frac{1}{0}$.

Prior information about frequencies of structural attributes (e.g., tree depth) or content attributes (e.g., operators) can be integrated as *priors* during both the structure and node value sampling processes. Structure priors modulate the sampling probabilities of specific structures. For example, the sampling probability of the tree structure in Figure 1 could be fixed, represented as $p(\text{structure} = (0, 1, 1, 2, 2))$, $p(\text{structure} = (0, 1, 2, 2, 3, 3))$, and $p(\text{structure} = (0, 1, 1, 2, 3, 4, 3))$. Beyond this direct probability manipulation, our sampler implementation can derive structure priors from broader structural characteristics, such as node count or tree depth.

After the structure is sampled, operators, functions, and features are sampled independently. Once again, the integration of priors is possible. For instance, one can modulate the sampling probabilities of specific operators such as $p(\text{operator node} = +)$, $p(\text{operator node} = -)$, and $p(\text{operator node} = *)$. With features, it is possible to tweak the likelihood of sampling either a variable or a constant, represented as $p(\text{leaf} = \text{variable})$ and $p(\text{leaf} = \text{constant})$. The sampler also incorporates conditional priors based on the parent node's value. This means the probability of an addition operator within a cosine can be adjusted as $p(\text{operator node} = \text{addition}|\text{parent} = \cos)$, and so on.[2]

---

[2]To simplify expressions and prevent complexities, we fix the probability of encountering a constant within a function to zero: $p(\text{leaf} = \text{constant}|\text{parent} = \text{function}) = 0$. This is motivated by cases in which constants within functions can invariably be reduced to a straight constant. For instance, $cos(0) = 1$ and $log(10) = 1$.

Given various post-sampling modifications (e.g., expression simplification), it is essential to re-calibrate the sampling probabilities of all the steps described above—structure sampling, operator sampling, function sampling, and feature sampling. To do so, we *burn* expressions by (a) sampling them, (b) analyzing the frequencies of expression attributes, and (c) fine-tuning the sampling probabilities. See Appendix A.2 for a comprehensive overview of this process.

In the last step before benchmarking, the sampler can sample an arbitrary amount of experimental conditions and produce data points while introducing different noise levels to the dependent variable (see Appendix A.3.2).

## 3 Experiment

One of the key advantages of utilizing a dynamic equation benchmark lies in its capacity for a nuanced, systematic exploration of the factors that influence the efficacy of SR algorithms. In this experiment, we examine how varying levels of equation complexity impact the performance of four SR algorithms.

### 3.1 Methods

#### 3.1.1 Symbolic Regression Algorithms

We evaluate the performance of four of the best-performing SR algorithms from SRBench: gplearn [3], AIFeynman [2], GP-GOMEA [26], [27], SBP-GP [28].

*gplearn.* The most traditional implementation of GP-based SR algorithm we tested is gplearn, which initializes a random population of models, then iterates through tournament selection, mutation, and crossover.

*AIFeynman.* AIFeynman is a divide-and-conquer algorithm that builds symbolic models [2]. It employs brute-force solvers and problem-decomposition techniques, including differentiated function approximations. Here, we probed the most recent version of this method that combines Pareto optimization with an information-theoretic complexity metric [29] to yield simple equations.

*GP-GOMEA.* GP-GOMEA is an extension of the GOMEA evolutionary algorithm to GP, designed to efficiently mix and preserve beneficial tree structures in the population [26], [27].

*SBP.* Semantic backpropagation (SBP) computes a value at a tree node position to align the model's output with the target [30]–[32]. We assess the GP SBP-GP algorithm[28], which enhances SBP-based recombination through dynamic adjustments of intermediate outputs with affine transformations.

#### 3.1.2 Datasets

We based all analyses reported below on an equation dataset resembling equations from the domain of physics. We first extracted equation priors based on the frequencies of tree structures, input variables, constants, operators, and functions of physics equations scraped from Wikipedia (see Appendix A.3.4). We then generated 120 equations using those priors. Finally, we randomly sampled 1000 data points representing the input variables and output of the respective ground-truth equation. The input variables were sampled from the range [-10, 10].

### 3.2 Metrics

*Independent Variables: Complexity Metrics.* Our primary interest lies in the effects of different complexity metrics on the performance of SR algorithms. Here, we focus on the following metrics extracted from each equation: (1) number of nodes, (2) expression tree depth, (3) number of input variables, (4) number of constants, and (5) number of trigonometric functions[4].

*Dependent Variables: Performance Metrics.* In the first analysis, we examine the impact of each complexity metric on three performance measures commonly used to evaluate SR algorithms: symbolic

---

[3]Documentation can be found here: `https://gplearn.readthedocs.io/en/stable/`

[4]The number of trigonometric functions was used as complexity metric since it had been shown to impact the performance in various SR algorithms

solution rate, normalized tree distance (NED), and mean square error (MSE). The symbolic solution rate indexes if the predicted equation matches the ground truth or if it can be derived by scaling or constant shifting [1]. The NED measures the distance between the tree representations of the ground truth and predicted equations in terms of the number of tree modifications needed to transform the latter into the former [33]. We compute the MSE between the predicted and the ground truth equation on a randomly sampled set of hold-out conditions. For more details on each performance metric, see Appendix A.3.3

## 3.3 Simulation Procedure

We formatted the generated datasets to be compatible with SRBench, and executed the benchmarking[5] on the four algorithms using the following hyperparameters: 1 recovery trial allowed for each algorithm per equation, 16384 MB memory limit, nine hours time limit, and allowing for estimators to be tuned on black-box regression problems. After obtaining a symbolic solution, we computed all metrics for the obtained symbolic model, including symbolic solution and MSE. We leveraged the expression tree representation described above to compute the complexity metrics and the NED.

## 3.4 Analysis

We first examine the degree to which each SR performance metric is impacted by (a) the SR algorithm itself and (b) the complexity metric. Thus, for each complexity metric, we regress the respective performance metric against the complexity metric and the algorithms, using either a logistic regression (for symbolic solution hit) or linear regression (for NED and MSE). An interaction effect between the complexity metric and algorithms would indicate that different algorithms are differentially impacted by the different complexity metrics.

In the second analysis, we examine whether a given SR algorithm is capable of recovering the complexity metric of the ground-truth equation, e.g., whether the output equation of an SR algorithm had as many constants as the data-generating equation. If a feature is recovered perfectly, the regression should be an identity. Regression lines above the identity lines suggest overly complicated expressions in this feature, and regression lines below the identity lines suggest oversimplification of this feature.

## 3.5 Results

Table 1: Main effects of algorithm and complexity metrics for the logistic (symbolic solution probability) and linear regression (NED and MSE).

|  | Symbolic Solution | | | NED | | | MSE | | |
|---|---|---|---|---|---|---|---|---|---|
|  | OR | CI | p | coef | CI | p | coef | CI | p |
| Intercept | .88 | [.66, 1.17] | .377 | .42 | [.37, .47] | **<.001** | .77 | [.12, 1.43] | **.02** |
| AIFeynman | 1.57 | [1.04, 2.35] | **.03** | -.13 | [-.2, -.07] | **<.001** | -.23 | [-1.13, .68] | .625 |
| GP-GOMEA | .74 | [.5, 1.09] | .124 | .13 | [.07, .2] | **<.001** | .46 | [-.43, 1.35] | .308 |
| SBP-GP | .69 | [.46, 1.04] | .073 | .3 | [.23, .36] | **<.001** | -.21 | [-1.15, .74] | .667 |
| Nodes | .54 | [.49, .6] | **<.001** | .04 | [.03, .04] | **<.001** | .27 | [.18, .37] | **<.001** |
| Depth | .2 | [.16, .25] | **<.001** | .13 | [.11, .14] | **<.001** | .89 | [.62, 1.15] | **<.001** |
| Variables | .33 | [.27, .4] | **<.001** | .12 | [.11, .14] | **<.001** | 1.34 | [1.03, 1.65] | **<.001** |
| Constants | .35 | [.29, .43] | **<.001** | .11 | [.09, .14] | **<.001** | .02 | [-.31, .36] | .886 |
| Trigonometric | .15 | [.06, .36] | **<.001** | .29 | [.18, .4] | **<.001** | .31 | [.18, .44] | **<.001** |

*Note.* Odds ratio (OR) for logistic regression and coefficient (coef) for linear regression. Confidence interval (CI), p-value (p) for both.

*Main Effects.* Figure 2 illustrates the symbolic solution probability as a function of various complexity metrics for different algorithms, highlighting the main effects of algorithms and the decrease in performance for all algorithms as the complexity increases (for the other performance metrics, see Appendix A.4). Table 1 shows these main effects of the SR algorithm and the complexity metrics on all the performance metrics, the symbolic solution, NED, and MSE. For example, AIFeynman was significantly more likely to find a symbolic solution than the baseline (gplearn), and also yielded

---

[5]`https://github.com/cavalab/srbench/blob/master/experiment/analyze.py`

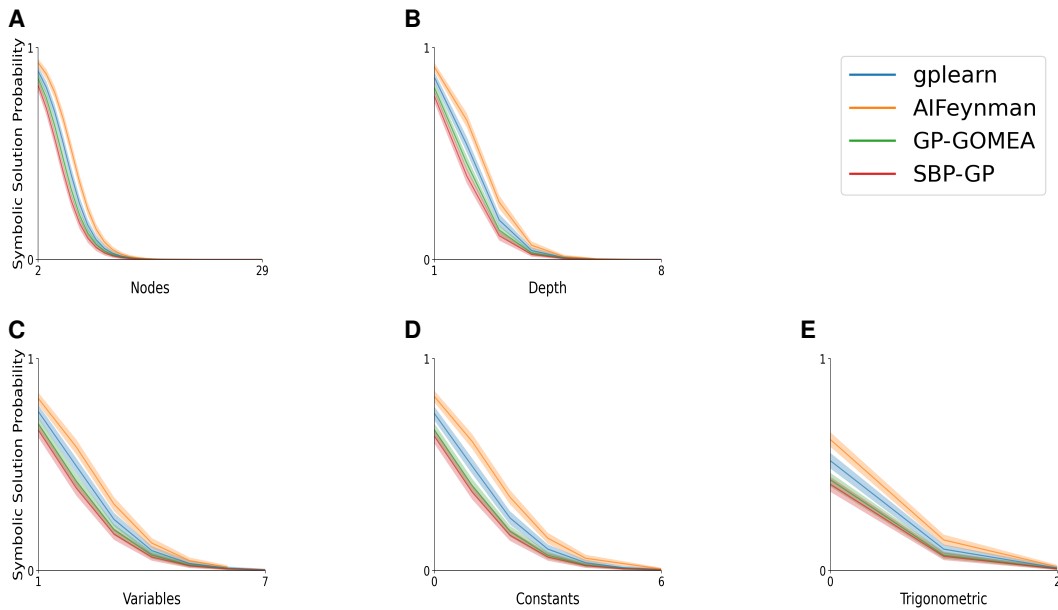

Figure 2: Symbolic solution probability as a function of various complexity metrics for different algorithms. On all the complexity metrics, the performance of AIFeynman is best, followed by gplearn, GP-GOMEA, and SBP-GP.

a significantly smaller NED between the ground truth and the proposed equation was smaller. GP-GOMEA and SBP-GP had a higher NED between the ground truth and the proposed equation compared to gplearn. None of the algorithms differed significantly in terms of MSE. Together, these results indicate that the SR algorithms impact performance metrics differently. All of the main effects of the complexity metrics except for Constants on MSE were significant. The higher the value of each complexity was, the lower the algorithm's performance on each performance metric was.

Table 2: Interaction effects of logistic (symbolic solution probability) and linear regression (NED and MSE) against algorithms and various complexity metrics.

| | Symbolic Solution | | | NED | | | MSE | | |
|---|---|---|---|---|---|---|---|---|---|
| | OR | CI | p | coef | CI | p | coef | CI | p |
| AIFeynman:Nodes | 1.26 | [.82, 1.92] | .293 | .02 | [.0, .03] | **.01** | .06 | [-.21, .33] | .659 |
| GP-GOMEA:Nodes | 1.74 | [1.19, 2.55] | **.004** | -.03 | [-.04, -.02] | **<.001** | .08 | [-.16, .31] | .521 |
| SBP-GP:Nodes | 2.31 | [1.61, 3.32] | **<.001** | -.02 | [-.03, -.01] | **<.001** | -.03 | [-.31, .25] | .833 |
| AIFeynman:Depth | 2.78 | [.99, 7.84] | .053 | .0 | [-.04, .04] | .988 | .03 | [-.71, .77] | .936 |
| GP-GOMEA:Depth | 3.36 | [1.24, 9.12] | **.018** | -.09 | [-.13, -.06] | **<.001** | .29 | [-.4, .98] | .407 |
| SBP-GP:Depth | 6.64 | [2.54, 17.35] | **<.001** | -.08 | [-.12, -.04] | **<.001** | -.0 | [-.84, .83] | .994 |
| AIFeynman:Variables | .72 | [.37, 1.4] | .329 | .04 | [-.01, .1] | .089 | -.02 | [-.97, .92] | .963 |
| GP-GOMEA:Variables | 1.19 | [.68, 2.09] | .547 | -.06 | [-.1, -.02] | **.003** | .41 | [-.4, 1.23] | .318 |
| SBP-GP:Variables | 1.54 | [.89, 2.68] | .126 | -.05 | [-.09, -.0] | **.03** | -.33 | [-1.23, .57] | .47 |
| AIFeynman:Constants | 1.16 | [.56, 2.39] | .695 | .001 | [-.05, .06] | .867 | -.03 | [-.97, .91] | .952 |
| GP-GOMEA:Constants | 1.96 | [1.02, 3.77] | **.043** | -.12 | [-.17, -.06] | **<.001** | -.05 | [-.95, .85] | .917 |
| SBP-GP:Constants | 3.03 | [1.6, 5.71] | **.001** | -.13 | [-.18, -.07] | **<.001** | -.14 | [-1.15, .87] | .784 |
| AIFeynman:Trigonometric | 2.31 | [.36, 14.97] | .381 | -.01 | [-.23, .2] | .895 | -.22 | [-.48, .04] | .096 |

*Note.* Odds ratio (OR) for logistic regression and coefficient (coef) for linear regression. Confidence interval (CI), p-value (p) for both.

*Interaction Effects.* Table 2 lists interaction effects between the algorithm and complexity metrics. Results indicate that different complexity metrics differently impacted different algorithms. This effect was most prevalent within the NED performance metric. For example, the interaction term for AIFeynman and number of nodes is significant. For every node, the average NED increase compared to gplearn is .02 units more. This means that the NED increases more rapidly as a function of nodes for AIFeynman, likely due to its simplicity bias. The interaction term for GP-GOMEA and node

count is also significant. For each node, the average change in the NED is .03 units less compared to gplearn. This indicates that the relationship between NED and complexity differs for the GP-GOMEA group, with the NED increasing less steeply. Together, the significant interaction effects show that the different complexity metrics differentially impacts different algorithms, with some algorithms (e.g., GP-GOMEA) scaling better with equation complexity (e.g., node count) compared to others (e.g., AIFeynman).

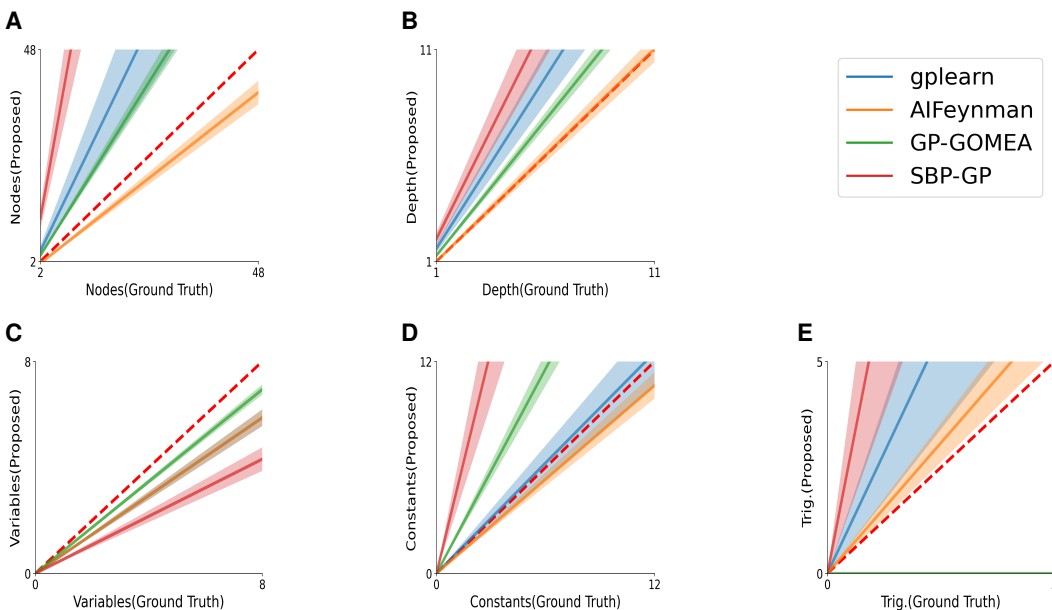

Figure 3: Recovered versus ground-truth equation complexity. Each figure depicts the linear regression fit obtained from regressing the equation complexity metric of the recovered equation against that of the ground-truth equation.

Figure 3 highlights the degree to which the recovered and ground-truth equations match one another in terms of different complexity metrics. Table 3 shows the deviation of the regression line between the proposed and the ground truth metrics from the identity. We observed that the number of nodes was higher in the predicted equations for all algorithms other than the AIFeynman algorithm (which predicted a lowe node number). The tree depth was recovered well by AIFeynman and overestimated by the other algorithms. The number of input variables was underestimated by all of the algorithms. The number of constants was overestimated by GP-GOMEA and SBP-GP and recovered well by gplearn and AIFeynman. The number of trigonometric functions was overestimated by all but GP-GOMEA, which did not propose a single trigonometric function even when they were present in the ground truth equation.

Table 3: Linear regression of various proposed complexity metrics against the ground truth metrics. In this analysis, the identity was subtracted from the proposed equation metrics. Values deemed significant indicate that the regression line deviates from the identity line.

| Metric | gplearn | | | AIFeynman | | | GP-GOMEA | | | SBP-GP | | |
|---|---|---|---|---|---|---|---|---|---|---|---|---|
| | coef | CI | p | coef | CI | p | coef | CI | p | coef | CI | p |
| Nodes | 1.14 | [.58, 1.69] | **<.001** | -.19 | [-.24, -.14] | **<.001** | .64 | .55, .74 | **<.001** | 4.7 | [3.54, 5.86] | **<.001** |
| Depth | .61 | [.41, .82] | **<.001** | .003 | [-.05, .06] | .91 | .28 | [.22, .34] | **<.001** | 1.06 | [.77, 1.36] | **<.001** |
| Var. | -.26 | [-.30, -.23] | **<.001** | -.26 | [.31, -.22] | **<.001** | -.13 | [-.16, -.11] | **<.001** | -.46 | [-.52, -.41] | **<.001** |
| Const. | .04 | [-.12, .21] | .6 | -.11 | [-.18, -.05] | **<.001** | .93 | [.77, 1.1] | **<.001** | 3.21 | [2.2, 4.23] | **<.001** |
| Trig. | 1.28 | [.36, 2.19] | **.007** | .23 | [.05, .41] | **<.001** | | | | 4.48 | [2.01, 6.95] | **<.001** |

*Note.* Coefficient (coef), confidence interval (CI), p-value (p) for linear regression.

# 4 Conclusion and Future Work

We introduced a new benchmarking approach for SR. In contrast to the existing datasets for benchmarking SR algorithms, the introduced benchmarking method relies on an expression sampler capable of generating an arbitrary number of expressions. Critically, the user may generate those expressions based on user-defined metrics of equation complexity, such as a number of expression tree nodes, expression tree depth, or the number of variables, or based on priors for a specific domain (e.g., physics). Using this sampler, we examined the performance of various SR methods based on different equation complexity metrics. We found that the performance of the SR algorithms decreases with equation complexity. In particular, we found that the degree to which the recovered and ground-truth equations match one another as a function of different complexity metrics varies, with tree depth being most often recovered and trigonometric functions least often.

Our results suggest that the benchmarking method can yield novel insights into the strengths and weaknesses of existing SR algorithms, such as which kinds of equation complexities different SR algorithms fail to capture. Insights into complexity-specific weaknesses may not just aid in the comparison of existing SR algorithms but may also help steer the development of novel SR algorithms. For instance, we observed that most SR algorithms we evaluated (except AIFeynman) overestimate the complexity of an equation in terms of the number of expression tree nodes or the number of trigonometric functions. This suggests that SR algorithms may benefit from biases against such complexity.

A crucial yet unexplored feature of our benchmarking method pertains to the ability to sample equations according to domain-specific priors. Existing (static) benchmarks for equations are often aligned with problems in physics, potentially tempting SR researchers to tailor their algorithms toward these datasets. This may come at the expense of generalizability across other types of equations and scientific domains. For instance, in biology, equations may tend to incorporate more exponential functions to model complex growth phenomena, while in physics, multiplications might dominate equations describing physical laws. Future benchmarking efforts may benefit from evaluating the domain-specificity of different SR algorithms by examining their performance across equation datasets generated from priors based on different scientific disciplines. Thus, our novel benchmarking approach not only offers a more flexible and comprehensive evaluation of SR algorithms but also highlights the need for future research to consider domain-specificity, thereby enhancing the generalizability and adaptability of these algorithms across various scientific disciplines.

## Acknowledgments and Disclosure of Funding

This project is in active development by the Autonomous Empirical Research Group. The development of this package is supported by Schmidt Science Fellows, in partnership with the Rhodes Trust, as well as the Carney BRAINSTORM program at Brown University.

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

# A Appendix

## A.1 Expression Tree

We represent mathematical expressions as trees with operators or functions as internal nodes and constants or variables as leaves. For this approach to be suitable for ML and especially SR benchmarking, we introduce conventions to make the trees identifiable and the mapping between string representation and tree a one-to-one correspondence:

**Nodes can have a maximum of two children.** Expressions like sums and products may correspond to several trees. For instance, the expression $x_1 + x_2 + 2$ can be represented as any of various trees as shown in Figure A1. Here, we assume that all operators have at most two operands, resulting in an expression tree data structure where each node can have one or two children.

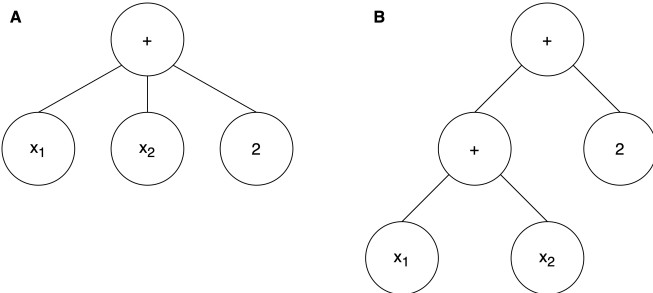

Figure A1: The mathematical expression $x_1 + x_2 + 2$ using two distinct equation tree structures: Tree (A), characterized by nodes with an arbitrary number of children, and Tree (B), featuring nodes with a maximum of two children.

**Trees use binary minus and division.** In many mathematical frameworks, the operation of subtraction is commonly represented as the addition of an additive inverse:

$$x - y := x + (-y) \text{ where } -y := y' \text{ with } y + y' = 0$$

The same holds for division. It is represented as the multiplication of a multiplicative inverse :

$$x/y := x * y^{-1} \text{ where } y^{-1} := y' \text{ with and } y * y^{-1} = 1 \ (\forall \ y \neq 0)$$

While the conventional notation benefits from the commutative property of operators, our proposed notation introduces distinctions by exclusively allowing binary minus and division operations. This notation aligns more closely with the prevalent usage in (SR) algorithms and offers the advantage of generally generating shallower tree structures (see Figure A2). When required, unary minus is transformed into a binary minus following the convention that $-x$ is replaced with $0 - x$ if converting a plus sign to a minus sign at another location is not feasible.

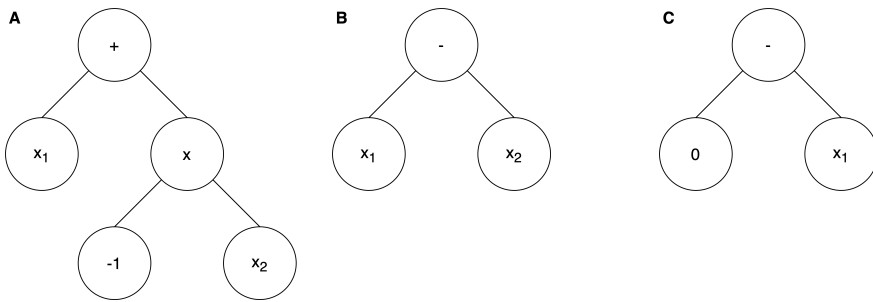

Figure A2: Expressions with unary or binary minus: Tree (A), characterized by the unary minus, representing the additional inverse. Tree (B), featuring the same expression as (A) with a binary minus. Tree (C) expressing $-x_1$ with a binary minus.

## A.2 Expression Sampler

In our expression sampling process, we simplify expressions and remove invalid ones. For instance, after creating a tree structure and populating its nodes, we might produce expressions such as $\sqrt{x^2}$ or $\sqrt{-|x+1|}$. While the former reduces to $x$, the latter is eliminated because it lacks a valid domain in $\mathbb{R}$.

This post-sampling adjustment can cause a disparity between the intended sampling probabilities and the actual outcomes since the likelihood of an expression being simplified or changed isn't independent of its structural and mathematical elements. For example, expressions containing log, power, or $\sqrt{}$ functions are more prone to being discarded.

Users can perform an *expression burn-in* procedure to address this discrepancy. Here, expressions are sampled, and their attribute frequencies are measured against the desired attribute probabilities. If there's a misalignment, an adjusted probability is recorded and subsequently used for sampling, steering the results closer to the original goal.

Our repository already includes pre-calculated adjusted probabilities for certain target distributions, like uniform sampling. Moreover, the burn-in procedure only needs to be executed once. Once completed, these adjusted probabilities are saved to a file for future reference.

## A.3 Additional Features

### A.3.1 Collection of Priors

The expression sampler can collect information about frequencies of given sets of equations—for example, from existing benchmarking datasets. This information can then be used to inform the sampling of expressions.

### A.3.2 Sampling of Datapoints

The expression sampler is tailored for SR benchmarking, allowing to export sampled expressions compatible with the SR Bench format [1]. The process begins by sampling a user-specified number of conditions. Subsequently, the sampled expression is evaluated against these conditions, from which we generate data tables and an associated meta-data file.

### A.3.3 Benchmarking Metrics

Our expression sampler includes various metrics to measure the 'accuracy' of the SR algorithm's output against the sampled expressions.

*Prediction distance.* Among these is the prediction distance proposed by Cava, Orzechowski, Burlacu, *et al.* [1]:

$$d_{prediction} = \sum_{i=1}^{N} (f_{pred}(X_i) - f_{true}(X_i))^2 / N$$

where N indicates the number of samples.

*Symbolic solution.* Another metric proposed by Cava, Orzechowski, Burlacu, *et al.* [1] is called symbolic solution, designed to capture SR models that differ from the true model by a constant or scalar. In our application, we define the symbolic constant difference as:

$$d_{constant} = \begin{cases} c & \exists c \in \mathbb{R} : (f_{pred} - f_{true})(X) = c \\ \infty & otherwise \end{cases}$$

And the symbolic scalar difference as:

$$d_{scalar} = \begin{cases} s & f_{true} \neq 0 \text{ and } \exists s \in \mathbb{R} : (f_{pred}/f_{true})(X) = s \\ \infty & otherwise \end{cases}$$

*Normalized edit distance.* In addition to the metrics above, Matsubara, Chiba, Igarashi, *et al.* [33] propose a normalized edit distance for the trees. Its primary use has been to study the search process for genetic programming approaches [34]–[36]. For a pair of two trees, edit distance computes the minimum cost to transform one to another with a sequence of operations, each of which either 1) inserts, 2) deletes, or 3) renames a node. To calculate the tree edit distance, we use the algorithm proposed by Zhang and Shasha [37] and normalize it via the following equation:

$$ned(t_1, t_2) = \frac{2 * ed(t_1, t_2)}{ed(t_1, t_2) + |t_1| + |t_2|},$$

where $ned(t_1, t_2)$ is the normalized edit distance between two trees $t_1$ and $t_2$. This normalization method has the advantage over naive normalization methods like $\frac{ed(t_1,t_2)}{|t_1|+|t_2|}$ or $\frac{ed(t_1,t_2)}{max(|t_1|,|t_2|)}$ that it satisfies the triangle inequality $(ned(t_1, t_2) + ned(t_2, t_3) \geq ned(t_1, t_2) \forall t_1, t_2, t_3)$. It also satisfies the following axioms:

$$\text{Non-Negativity: } ned(t_1, t_2) \geq 0 \text{ and } ned(t_1, t_2) = 0 \Leftrightarrow t_1 = t_2$$
$$\text{Symetry: } ned(t_1, t_2) = ned(t_2, t_1)$$

It is, therefore, a metric. For a detailed proof, we refer readers to Li and Chenguang [38].

### A.3.4   Scraping Priors

The expression sampler can make use of priors. For example, to generate expressions to a specific scientific domain by incorporating domain-specific. The prior distribution contains the number of times each tree structure, operator, function, and feature appeared per equation.

We created an informed prior for the domain of Physics by webscraping equations from Wikipedia pages using the open-source package *equation scraper*[6]. This equation scraper accumulates equations from links to a certain depth and then parses scraped expressions using the expression tree. Here, we used a search depth. This means all links within the corresponding category page, links within these links, and finally, links within these sublinks were considered. The first path of the Physics domain was . We then extracted equations from all levels of these links for parsing.

### A.4   Additional Figures: Performance Metrics

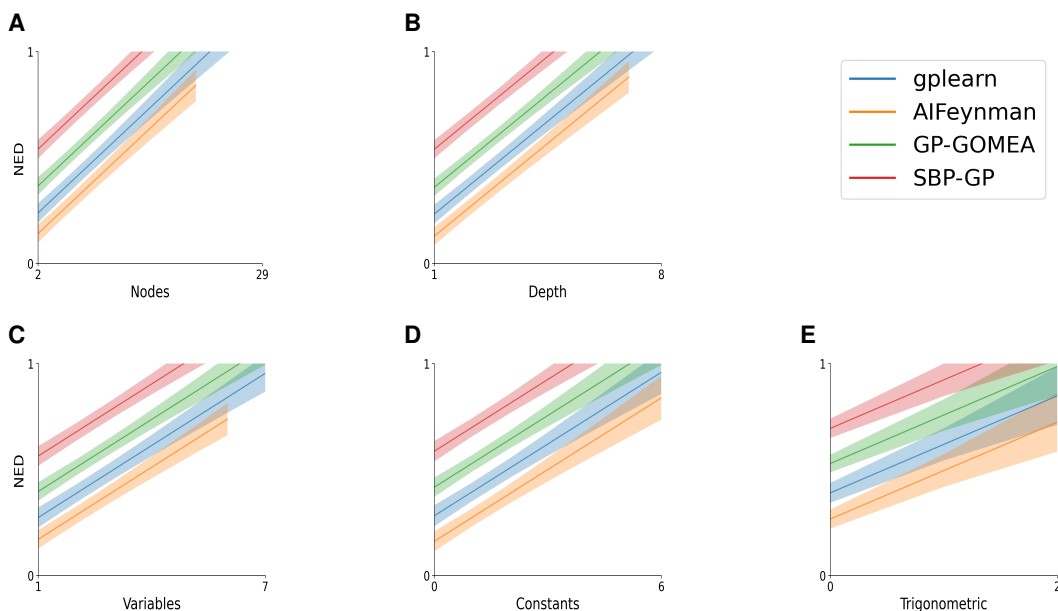

Figure A3: Normalized Edit Distance (NED) as a function of various complexity metrics for different algorithms and datasets. The dataset is only from sampled equations (Feynman Priors and Physics Priors). On all the complexity metrics, the performance of AIFeynman is best, followed by gplearn, GP-GOMEA, and SBP-GP.

---

[6]The scraper is available as pyhthon package and documented athttps://autoresearch.github.io/equation-scraper/.

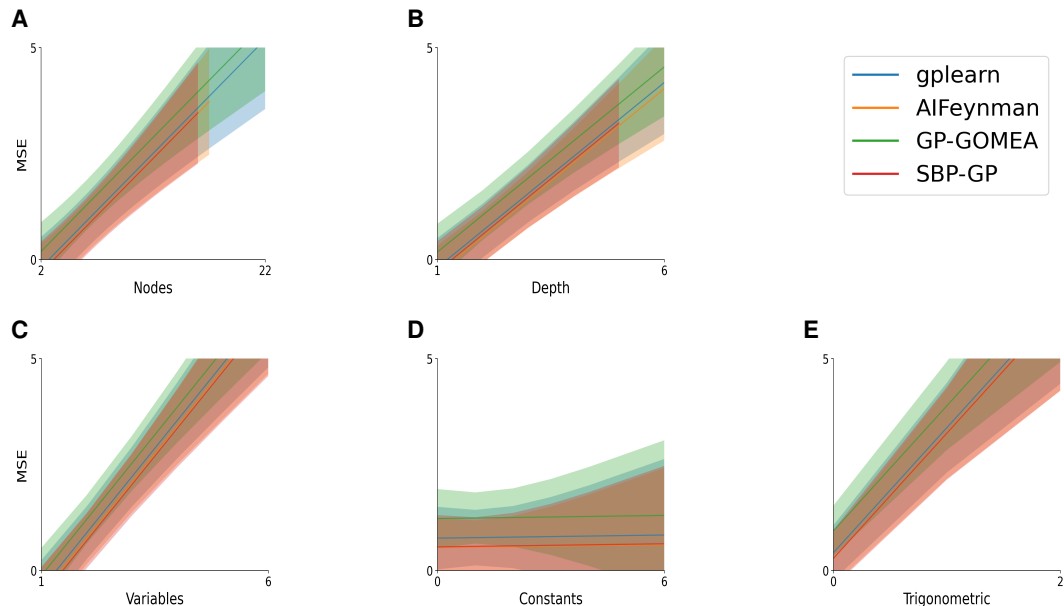

Figure A4: Mean Square Error (MSE) as a function of various complexity metrics for different algorithms and datasets. The dataset is only from sampled equations (Feynman Priors and Physics Priors). On all the complexity metrics, the performance of AIFeynman is best, followed by gplearn, GP-GOMEA, and SBP-GP.

## A.5   Sampled Expressions

A1 shows the benchmark expressions we used. Each constant $c_i$ was sampled uniformly between 0 and 5. We sampled with replacement so that an expression can occur more than once in the set.[7]

---

[7]The expression sampler also allows for sampling without replacement

Table A1: Sampled expressions: Wikipedia Physics prior

| Expression | Variables | Constants | Structure | Nodes | Trigonometric Functions |
|---|---|---|---|---|---|
| $x_1 x_2$ | 2 | 0 | [0, 1, 1] | 3 | 0 |
| $x_1 x_2$ | 2 | 0 | [0, 1, 1] | 3 | 0 |
| $x_1^2 x_2 x_3 x_4$ | 4 | 1 | [0, 1, 2, 3, 4, 4, 3, 2, 1] | 9 | 0 |
| $x_1^4$ | 1 | 1 | [0, 1, 1] | 3 | 0 |
| $-c_0 + x_1^2$ | 1 | 2 | [0, 1, 2, 2, 1] | 5 | 0 |
| $x_1 x_2$ | 2 | 0 | [0, 1, 1] | 3 | 0 |
| $x_1^{x_2}$ | 2 | 0 | [0, 1, 1] | 3 | 0 |
| $x_1 x_2$ | 2 | 0 | [0, 1, 1] | 3 | 0 |
| $c_0 x_1 x_2$ | 2 | 1 | [0, 1, 2, 2, 1] | 5 | 0 |
| $x_1 x_2$ | 2 | 0 | [0, 1, 1] | 3 | 0 |
| $\sqrt{x_1 x_2}$ | 1 | 0 | [0, 1, 2, 2] | 4 | 0 |
| $c_0 x_1^4$ | 1 | 1 | [0, 1, 2, 3, 3] | 5 | 0 |
| $c_0 + x_1$ | 1 | 1 | [0, 1, 1] | 3 | 0 |
| $x_2^3 \left(-c_0 + x_1 + x_2\right)^2$ | 3 | 1 | [0, 1, 2, 3, 4, 4, 3, 2] | 8 | 0 |
| $x_1^4$ | 1 | 1 | [0, 1, 1] | 3 | 0 |
| $c_0 x_1$ | 1 | 1 | [0, 1, 1] | 3 | 0 |
| $x_1 x_2 x_3$ | 3 | 0 | [0, 1, 2, 2, 1] | 5 | 0 |
| $c_0 x_1$ | 1 | 1 | [0, 1, 1] | 3 | 0 |
| $c_0^{x_3} x_1 x_2$ | 3 | 1 | [0, 1, 2, 3, 3, 2, 1] | 7 | 0 |
| $x_1^2 x_2^2 x_3$ | 3 | 0 | [0, 1, 2, 3, 3, 2] | 6 | 0 |
| $x_1^2$ | 2 | 1 | [0, 1, 1] | 3 | 0 |
| $c_0 + x_1 x_2 + x_1$ | 2 | 1 | [0, 1, 2, 2, 3, 3, 1] | 7 | 0 |
| $x_1^4$ | 1 | 1 | [0, 1, 1] | 3 | 0 |
| $2 x_1$ | 1 | 0 | [0, 1, 1] | 3 | 0 |
| $c_0 x_1$ | 1 | 1 | [0, 1, 1] | 3 | 0 |
| $x_1 + x_2$ | 2 | 1 | [0, 1, 1] | 3 | 0 |
| $(c_0 + x_1)^2$ | 1 | 0 | [0, 1, 2, 2] | 4 | 0 |
| $c_0 x_2^2$ | 1 | 1 | [0, 1, 2, 2] | 4 | 0 |
| $c_0 x_1^6 x_2$ | 2 | 1 | [0, 1, 2, 3, 4, 4, 3] | 7 | 0 |
| $c_0 x_1$ | 1 | 1 | [0, 1, 1] | 3 | 0 |
| $c_0 x_1 + c_1$ | 1 | 2 | [0, 1, 2, 2, 1] | 5 | 0 |
| $-c_0 + x_1 x_2$ | 2 | 1 | [0, 1, 2, 2, 1] | 5 | 0 |
| $\frac{x_1 x_3}{x_2^3}$ | 3 | 0 | [0, 1, 1, 2, 2] | 5 | 0 |
| $c_0 \sqrt{x_1 x_2}$ | 2 | 0 | [0, 1, 2, 2] | 4 | 0 |
| $\frac{x_2^2}{x_2^2}$ | 2 | 1 | [0, 1, 2, 3, 3, 2] | 6 | 0 |
| $x_1^2$ | 2 | 0 | [0, 1, 2, 2] | 4 | 0 |
| $c_0 x_2^2$ | 1 | 1 | [0, 1, 1] | 3 | 0 |
| $x_1 x_2$ | 2 | 1 | [0, 1, 2, 2] | 4 | 0 |
| $x_1 x_2$ | 1 | 0 | [0, 1, 1] | 3 | 0 |
| $c_0 - x_1 x_2 x_3$ | 3 | 1 | [0, 1, 1, 2, 3, 3, 2] | 7 | 0 |
| $c_0 x_1 + x_2^2 x_3^2 - \sin\left(c_1 + x_2\right)$ | 3 | 2 | [0, 1, 2, 3, 3, 2, 3, 4, 4, 1, 2, 3, 3] | 13 | 1 |
| $\frac{x_1 x_2}{c_0 x_2}$ | 2 | 0 | [0, 1, 1] | 3 | 0 |
| $\frac{x_1^2}{x_2^2}$ | 2 | 1 | [0, 1, 1, 2, 2] | 5 | 0 |
| $c_0 x_1^2 x_2^2$ | 2 | 0 | [0, 1, 2, 2, 3, 3] | 6 | 0 |
| $x_1 x_2$ | 2 | 0 | [0, 1, 1] | 3 | 0 |
| $c_0 x_2^3 x_3 \left(c_1 x_1 - 1\right)^3$ | 3 | 3 | [0, 1, 2, 3, 4, 5, 5, 4, 3, 4, 4, 2] | 12 | 0 |
| $\left(\frac{x_1}{x_2}\right)^{\frac{x_3}{x_4}}$ | 4 | 0 | [0, 1, 2, 2, 1, 2, 2] | 7 | 0 |

| Expression | | | | | |
|---|---|---|---|---|---|
| $c_0 x_1^2$ | 1 | 1 | [0, 1, 2, 2] | 4 | 0 |
| $\frac{c_0 x_1}{x_2}$ | 2 | 1 | [0, 1, 1, 2, 3, 3] | 6 | 0 |
| $x_1 x_2$ | 2 | 0 | [0, 1, 1] | 3 | 0 |
| $c_0 x_2$ | 1 | 1 | [0, 1, 1, 2] | 4 | 0 |
| $c_0 x_1 x_2^2$ | 2 | 0 | [0, 1, 2, 3, 4, 5, 5, 4] | 8 | 0 |
| $x_1 x_2$ | 2 | 0 | [0, 1, 1] | 3 | 0 |
| $x_1 x_2$ | 2 | 0 | [0, 1, 1] | 3 | 0 |
| $c_0 x_1$ | 1 | 1 | [0, 1, 1] | 3 | 0 |
| $(x_1 - x_2)^2$ | 2 | 0 | [0, 1, 2, 2] | 4 | 0 |
| $\frac{c_0}{x_1^3}$ | 1 | 1 | [0, 1, 2, 2] | 4 | 0 |
| $c_0 x_1^2$ | 1 | 1 | [0, 1, 2, 2] | 4 | 0 |
| $x_1 x_2$ | 2 | 0 | [0, 1, 1] | 3 | 0 |
| $c_1 - x_1 x_2 x_3 - c_0 x_4^{12} x_5^{12}$ | 5 | 2 | [0, 1, 1, 2, 3, 3, 2, 3, 3, 4, 5, 6, 7, 8, 8, 7] | 16 | 0 |
| $c_0 x_1$ | 1 | 1 | [0, 1, 1] | 3 | 0 |
| $(-c_0 + x_1)^2$ | 2 | 1 | [0, 1, 2, 2] | 4 | 0 |
| $x_2$ | 1 | 0 | [0, 1, 1] | 3 | 0 |
| $c_0 x_1$ | 1 | 1 | [0, 1, 1] | 3 | 0 |
| $x_1^4$ | 1 | 0 | [0, 1, 1] | 3 | 0 |
| $x_1 + x_2$ | 2 | 1 | [0, 1, 1] | 3 | 0 |
| $c_0 + x_1$ | 1 | 1 | [0, 1, 2, 2, 1] | 5 | 0 |
| $c_0 x_1 x_2$ | 2 | 2 | [0, 1, 1, 2, 2] | 5 | 0 |
| $c_0^4 x_1 - x_2^4$ | 2 | 1 | [0, 1, 2, 3, 4, 4, 3] | 7 | 0 |
| $x_1 c_0 x_2 x_2 + 4$ | 2 | 2 | [0, 1, 1, 2, 3, 3, 2] | 7 | 0 |
| $c_0 x_1$ | 2 | 0 | [0, 1, 1] | 3 | 0 |
| $x_1 x_2$ | 2 | 0 | [0, 1, 1] | 3 | 0 |
| $c_0 x_1$ | 1 | 0 | [0, 1, 1] | 3 | 0 |
| $x_1^2$ | 2 | 0 | [0, 1, 1] | 3 | 0 |
| $x_1 - x_2$ | 2 | 1 | [0, 1, 2, 3, 4, 4, 3, 2] | 8 | 0 |
| $x_1^2 (x_1 x_2)^{c_0}$ | 2 | 4 | [0, 1, 2, 3, 4, 5, 5, 4, 3, 2, 3, 3] | 12 | 0 |
| $c_0 c_1 (c_2 x_1 - 1)^{c_3}$ | 2 | 2 | [0, 1, 2, 2, 1, 2, 2] | 7 | 0 |
| $(c_0 + x_1)^3$ | 1 | 1 | [0, 1, 2, 2] | 4 | 0 |
| $\cos(c_0 - x_1)$ | 1 | 1 | [0, 1, 2, 2] | 4 | 1 |
| $c_0 x_1^2 x_2$ | 2 | 2 | [0, 1, 2, 3, 3, 1] | 7 | 0 |
| $x_1 x_2 x_3 x_2^2$ | 4 | 0 | [0, 1, 2, 3, 3, 4, 4, 2] | 8 | 0 |
| $-c_0 + x_1^4 x_2$ | 2 | 1 | [0, 1, 2, 2, 1] | 5 | 0 |
| $x_1 x_2$ | 2 | 0 | [0, 1, 1] | 3 | 0 |
| $x_1 e^{x_2}$ | 2 | 0 | [0, 1, 1, 2] | 4 | 0 |
| $\frac{c_0}{x_1^2 x_2}$ | 2 | 2 | [0, 1, 1, 2, 3, 4, 4, 3] | 8 | 0 |
| $x_1^4 x_2$ | 2 | 0 | [0, 1, 2, 3, 3] | 5 | 0 |
| $x_1 x_2$ | 2 | 0 | [0, 1, 1] | 3 | 0 |
| $x_1 x_2$ | 2 | 0 | [0, 1, 2, 2] | 4 | 0 |
| $x_1 x_2$ | 2 | 0 | [0, 1, 1] | 3 | 0 |
| $\frac{x_2}{x_3}$ | 3 | 0 | [0, 1, 1, 2, 2] | 5 | 0 |
| $\sin(c_0 - x_1)$ | 1 | 1 | [0, 1, 2, 2] | 4 | 1 |
| $2 x_1 x_2$ | 2 | 1 | [0, 1, 2, 2, 1] | 5 | 0 |

| | | | | | |
|---|---|---|---|---|---|
| $-c_0 + x_1$ | 1 | 1 | [0, 1, 1] | 3 | 0 |
| $x_1 x_3$ | 3 | 0 | [0, 1, 1, 2, 2] | 5 | 0 |
| $(-c_0 + x_1 x_2)^2$ | 2 | 1 | [0, 1, 2, 3, 3, 2] | 6 | 0 |
| $\frac{c_0 x_1 x_2}{c_0}$ | 2 | 1 | [0, 1, 2, 2, 1] | 5 | 0 |
| $e^{x_1}$ | 1 | 1 | [0, 1, 2, 2] | 4 | 0 |
| $x_1 x_2 x_3$ | 3 | 0 | [0, 1, 2, 2, 1] | 5 | 0 |
| $c_0 x_1^2$ | 1 | 1 | [0, 1, 2, 2] | 4 | 0 |
| $c_0 x_1$ | 1 | 1 | [0, 1, 1] | 3 | 0 |
| $c_0 x_1$ | 2 | 1 | [0, 1, 1] | 3 | 0 |
| $c_0 x_1 x_2$ | 1 | 1 | [0, 1, 2, 2, 1] | 5 | 0 |
| $c_0 x_1^4$ | 1 | 1 | [0, 1, 2, 3, 3, 2] | 6 | 0 |
| $c_0 x_1^2$ | 1 | 1 | [0, 1, 2, 2] | 4 | 0 |
| $(c_0 + x_1)^2$ | 1 | 2 | [0, 1, 1, 2, 2] | 5 | 0 |
| $c_0 x_1$ | 2 | 0 | [0, 1, 1] | 3 | 0 |
| $x_1 x_2$ | 1 | 1 | [0, 1, 2, 2] | 4 | 0 |
| $c_0 x_1^4$ | 2 | 0 | [0, 1, 2, 3, 3] | 5 | 0 |
| $x_1^4 x_2^4$ | 2 | 0 | [0, 1, 2, 2] | 4 | 0 |
| $\sqrt{x_1^9 + x_2}$ | 1 | 1 | [0, 1, 2, 3, 3] | 5 | 0 |
| $c_0 x_1$ | 1 | 1 | [0, 1, 1] | 3 | 0 |
| $c_0 x_1$ | 2 | 0 | [0, 1, 1] | 3 | 0 |
| $x_1 x_2$ | 2 | 2 | [0, 1, 2, 3, 4, 4, 3, 2] | 8 | 0 |
| $c_0 x_1^2 x_2^2$ | | | | | |

