# OpenReview forum: "Expression Sampler as a Dynamic Benchmark for Symbolic Regression"
_NeurIPS.cc/2023/Workshop/AI4Science — NeurIPS2023-AI4Science Poster_

### Official Review · Reviewer_Poah · 2023-10-16
**Interesting analysis for symbolic regression methods**

**Rating:** 7
**Confidence:** 4

**Review:**

This work introduces a new, dynamic benchmark framework for symbolic regression methods, using a symbolic expression sampler. While the reviewer finds it difficult to see how many data points are used to estimate trends for each of the methods like those in Figures 2 and 3, it seems like an interesting discussion and somewhat novel. Besides MSE, the analysis also involves recently proposed evaluation metrics such as symbolic solution (rate) and normalized edit distance.

This paper needs more clarifications and improvements in writing:
- Names should be consistent (Suggested vs. in this paper)
  - "AI Feynman" vs "AIFeynman"
  - "GP-GOMEA" vs "GPGOMEA"
- The reviewer could not understand structure formats like (0, 1, 1, 2, 2), which need clarifications.
- (NED, and MSE) -> (NED and MSE)
- Terms used in Tables 1 - 3 such as OR, CI, coef, p should be clarified
- "The higher the value of each complexity, the lower the algorithm’s performance on each performance metric." -> "The higher the value of each complexity is, the lower the algorithm’s performance on each performance metric is."
- "For example, The interaction" -> "For example, the interaction"
- "In a second analysis" -> "In the second analysis"
- Table A1 should be referenced and split into multiple sub-tables for better readability

The reviewer also suggests that the future work considers diverse SR problems with wider ranges of values to be sampled such as SRSD [19] and some normalized form of MSE as the magnitude of MSE value depends on the input/output value ranges, which are all fixed [-10, 10] in this study, and applying the same analysis with MSE to such diverse SR problems may be difficult.

---

### Official Review · Reviewer_pU4W · 2023-10-20
**A new framework to benchmark symbolic regression; can generate an arbitrary number of realistic tasks.**

**Rating:** 7
**Confidence:** 3

**Review:**

**Summary** The paper provides a new benchmark framework to evaluate and compare symbolic regression (SR) methods. This benchmark framework can generate new SR tasks based on priors learned from a limited set of real-world equations. Symbolic regression is a very relevant task for equation discovery and, thus, AI for science. The proposed framework extends existing benchmark approaches in a novel and meaningful direction.

*Strength*: The idea to learn domain-specific prior distributions from existing functions to create an arbitrary number of novel benchmark tasks related to real-world problems to overcome limitations of prior benchmark suites

*Weakness*: While the idea is very interesting, it is unclear to what extent the found solutions transfer to the original domain. It would've been interesting to see whether the drawn conclusions generalize to the original domain (from which the prior was derived).

*Note*: One of the main contributions of this work is a novel benchmark framework which needs to be accessible and reproducible by the community. Thus, the value of this contribution is closely related to the accessibility of the accompanying code. While the abstract mentions "we introduce an open-source method", I would highly encourage the authors to also add a link to their code in the Abstract/Paper for the final version.

Overall, I like the idea and (assuming that the code will be published) believe this is a valuable contribution to the AI4Science community.

Minor comments:
  * The readability of plots could be increased by adding more ticks or a grid in the background
  * Table 1/2: Abbreviations "OR" and "CI" are not introduced/explained. A brief description of the table content would help readers not familiar with the subject
  * Reference list would benefit from a second pass to clean/fix some entries, e.g. a unified format for preprints arxiv ([2] vs [3]), missing year ([8]), missing venue ([36]).

---

### Meta-Review · Area_Chair_SzDE · 2023-10-26

**Recommendation:** Accept (Poster)
**Confidence:** 3

**Metareview:**

This paper proposes a dynamic benchmark to compare symbolic regression methods. The reviewers are in agreement that the work is well motivated and the results are promising. They also raised a few concerns that should be taken seriously. Recommendation: Poster.